# Cold Brew Coffee—Pilot Studies on Definition, Extraction, Consumer Preference, Chemical Characterization and Microbiological Hazards

**DOI:** 10.3390/foods10040865

**Published:** 2021-04-15

**Authors:** Linda Claassen, Maximilian Rinderknecht, Theresa Porth, Julia Röhnisch, Hatice Yasemin Seren, Andreas Scharinger, Vera Gottstein, Daniela Noack, Steffen Schwarz, Gertrud Winkler, Dirk W. Lachenmeier

**Affiliations:** 1Chemisches und Veterinäruntersuchungsamt (CVUA) Karlsruhe, Weissenburger Strasse 3, 76187 Karlsruhe, Germany; lindaclaassen@gmx.de (L.C.); max.rinderknecht@gmx.de (M.R.); porth@rhrk.uni-kl.de (T.P.); julia.roehnisch@cvuaka.bwl.de (J.R.); haticeyasemin.seren@cvuaka.bwl.de (H.Y.S.); andreas.scharinger@cvuaka.bwl.de (A.S.); vera.gottstein@cvuaka.bwl.de (V.G.); daniela.noack@cvuaka.bwl.de (D.N.); 2Hochschule Albstadt-Sigmaringen, Fakultät Life Sciences, 72488 Sigmaringen, Germany; winkler@hs-albsig.de; 3Technische Universität Kaiserslautern, Erwin-Schrödinger-Straße, 67663 Kaiserslautern, Germany; 4Coffee Consulate, Hans-Thoma-Strasse 20, 68163 Mannheim, Germany; schwarz@coffee-consulate.com

**Keywords:** coffee, cold brew, nitro cold brew, roasting, extraction, hygiene, risk assessment, NMR, sensory analysis

## Abstract

Cold brew coffee is a new trend in the coffee industry. This paper presents pilot studies on several aspects of this beverage. Using an online survey, the current practices of cold brew coffee preparation were investigated, identifying a rather large variability with a preference for extraction of medium roasted Arabica coffee using 50–100 g/L at 8 °C for about 1 day. Sensory testing using ranking and triangle tests showed that cold brew may be preferred over iced coffee (cooled down hot extracted coffee). Extraction experiments under different conditions combined with nuclear magnetic resonance (NMR) analysis showed that the usual extraction time may be longer than necessary as most compounds are extracted within only a few hours, while increasing turbulence (e.g., using ultrasonication) and temperature may additionally increase the speed of extraction. NMR analysis also revealed a possible chemical differentiation between cold brew and hot brew using multivariate data analysis. Decreased extraction time and reduced storage times could be beneficial for cold brew product quality as microbiological analysis of commercial samples detected samples with spoilage organisms and contamination with *Bacillus cereus*.

## 1. Introduction

There are numerous ways to prepare a coffee beverage. For example, different extraction methods are used depending on personal preferences as well as the geographic, cultural and social context [1]. The extraction method influences the composition of the beverage [2]. Most common is the consumption of hot brewed coffee [3]. Here, the brewing process significantly influences the aroma of the coffee [3]. However, cold brew as a new extraction method has established itself on the market, and showed a 580% increase in sales in the US from 2011 to 2016 [4]. The new trend is replacing iced coffee more and more [5]. Iced coffee is hot brewed coffee which is then cooled [2].

However, cold brew is not a completely new concept. In the 17th century, this preparation method became known in Japan, when Dutchmen brought their coffee to Japan. One of the first cold brew system producers in the USA was Toddy, starting in the 1960s [5]. In 2010, cold brew finally gained popularity in the United States and four years later in the United Kingdom, and since 2016 it has also been known in Germany [6].

The sensory properties of cold brew depend on its production process. However, most studies regarding the sensory properties of coffee only refer to hot brewed coffee [7]. In contrast to the production of hot brewed coffee, there is still no uniform, standardized production process available for cold brew coffee with respect to parameters such as extraction time, extraction temperature, dosage, turbulence, water composition, bean type, grind and roasting [2]. There is a complete lack of definition of cold brew, not even what is meant by “cold”.

Although it might be self-evident that cold brew coffee is extracted with cold water, it is not yet specified how cold the water should be. The same applies to the extraction time, which is typically longer for cold brew than for hot brewed coffee. However, it is not clear how long the beverage should actually be extracted. Other parameters such as roasting, grind, turbulence and type of bean have also not yet been sufficiently researched.

Since there is still no optimized production process for cold brew coffee available, the aim of this work is to determine the influence of various parameters on cold brew coffee. Therefore, the extraction process, the current status of the manufacturing process and the preference for different cold brew coffees are investigated in this work by means of nuclear magnetic resonance (NMR) spectroscopy, as well as with a survey of cold brew manufacturers compounded with sensory and microbiological tests.

## 2. Materials and Methods

### 2.1. Online Survey on the Cold Brew Production Process

Using the SurveyMonkey tool (Survey Monkey Europe UC, Dublin, Ireland), a survey was created (see Kwok et al. [8] for details on development of the questionnaire and the data availability statement for access to a copy of the full questionnaire). This questionnaire contains questions which aim to find out how cold brew is produced at home, commercially and in the industry. This questionnaire was distributed on social networks with the collaboration of Coffee Consulate (Mannheim, Germany) and Earthlings Coffee Workshop (Kuching, Malaysia) and advertised during an international seminar on cold brew coffee (see Kwok et al. [8]) as well as on Facebook and Instagram, including non-public specialized interest groups on cold brew coffee. The participation of 125 people was achieved (see data availability statement for access to full raw results). The questionnaire results were evaluated using SurveyMonkey inbuilt statistical tools and Microsoft Excel 2016 V. 16.0 (Microsoft, Redmond, WA, USA).

### 2.2. Cold Brew Extraction Experiments

#### 2.2.1. Materials

Catuaí Arabica pulped natural processed beans (Fazendas Dutra—Specialty Coffee Production, São João do Manhuaçu, Brazil) were used for the production of all hot and cold brews in the experiments described here. Prior to the start of the experiments, 5 kg of the raw coffee beans were roasted using a Tyboon 3000 infrared roaster (Kammerer GmbH, Remchingen, Germany). These beans were carefully mixed through after roasting to ensure homogeneity. All following experiments were performed with these beans, unless otherwise stated, to exclude the influences of roasting, bean types, etc. on the results. The beans were freshly ground prior to the experiment using a Mahlkönig EK 43/1 coffee grinder (Mahlkönig, Hamburg-Wandsbek, Germany). Unless otherwise specified, grind grade 8 out of 12 was set.

#### 2.2.2. Production of Cold Brew Coffee

Experiment A: For the preparation of the cold brews, 260 ± 5 g beans are weighed in a beaker. The beans are then coarsely ground. The ground coffee powder is weighed again. Then, 240 ± 2 g are poured into a 5 L darkened glass container using a funnel. Three liters of water are now added, which has previously been tempered to 23 °C room temperature in a measuring beaker. After filling with water, the vessel is closed with a screw cap and shaken 5 times by holding the vessel upside down and then turning it upside down again. Immediately after this mixing process, the first sample is taken. This is done by pouring a small amount into a beaker via a funnel, which is fitted with a pleated filter. The sample is then transferred to a screw-top glass and stored in the refrigerator until measurement. The experiment is conducted in triplicate.

Experiment B: For this experiment, a total of three different cold brews were made from the same beans. For this purpose, 240 g of each of the roasted and ground beans were poured into 5 L amber glass bottles and 3 L of drinking water were added. All three bottles were shaken vigorously by hand after adding the water to ensure homogeneity of the coffee powder. All three extractions ran for 60 min with sampling every 5 min. One bottle was not moved except for sampling. A second bottle was placed in an ultrasonic bath and removed briefly only for sampling. A third bottle was locked on a shaking table with a movement frequency of 125 Hz and briefly removed only for sampling. For sampling, a small amount of the extract was tipped into a beaker via a funnel with a pleated filter. Subsequently, the filtered extract was transferred to 4 mL glass sample vials for storage at 5 °C using a glass pipette.

#### 2.2.3. Chemical Analysis by Means of NMR

Using NMR, the samples were analyzed for the content of formic acid, chlorogenic acid, caffeine, acetic acid, 5-hydroxymethylfurfural (HMF), lactic acid and trigonelline. A solution of 500 mg sodium trimethylsilylpropionate (TSP, CAS 37013-20-0) and 50 mL deuterium oxide (D_2_O, CAS 7789-20-0) was prepared as the internal reference standard for the NMR measurements. NMR buffer solution (pH 6.7) was prepared by mixing 500 mL H_2_O (dist.), 42 mL of KOH (1 M), 10 g of NaH_2_PO_4_ and 10 mg of sodium azide. Samples, buffer and internal standard were brought to room temperature before use.

For sample preparation, 100 μL buffer and 70 μL internal standard were first pipetted into a 4 mL glass screw vial using a 0.1 mL Eppendorf pipette. To the same screw vial, 600 μL of liquid coffee sample were added using a 1 mL Eppendorf pipette. The vials were then screwed shut and the contents mixed with a vortex mixer. After mixing, 600 μL of the solution were transferred to an NMR tube (Deutero Quant, glass, o.d. 4.966 ± 0.004 mm, i.d. 4.116 ±0.004 mm, length 17.78 cm) using a 1 mL Eppendorf pipette. The tube was then sealed and a barcode was applied. The spinner turbine was placed on top and the tube was wiped with a lint-free cloth with ethanol to remove any external contaminants. The sample tubes were placed in the SampleXpress H15040-01 sample changer of an Ultrashield 400 MHz NMR spectrometer with an Avance III console (Bruker, Rheinstetten, Germany). A 5 mm PASEI 1H/D-13C Z-GRD (Bruker, Rheinstetten, Germany) was used as the sample probe. The measurements were performed using a 1D-1H NOESY gppr pulse program with a gradient profile. The following parameters were defined: spectral width 20.55 ppm (8223.69 Hz), number of scans 32, dummy scans 4, acquisition time from 3.98 s, receiver gain of 16, dwell time from 60.80, delay (d1) of 4 s and a delay (d7) of 5 s. For an optimal pulse length, a calibration was performed for each measurement. The quantitative analysis took place via TopSpin software (Bruker, Rheinstetten, Germany) and using a MATLAB (Mathworks, Natick, MA, USA) script for automated integration (for details, see [9,10,11]). Graphs were generated using Prism 6 (GraphPad Software, San Diego, CA, USA).

### 2.3. NMR Differentiation between Cold Brew and Hot Brew Coffee

For exploratory data analysis of hot and cold brews, principal component analysis (PCA) was performed of the full NMR spectra of cold brew and hot brewed coffees analyzed as described in Section 2.2.3. The same Catuaí Arabica beans (see Section 2.2.1) were used to prepare hot and cold brews. The cold brews were prepared as described in Section 2.2.2. The hot brews were prepared using several different extraction methods, extraction degrees and extraction temperatures (88–96 °C), including the portafilter method, fully automatic coffee machines, French press, Karlsbad coffee maker, RS-16 glass filtration and hand filtration. Furthermore, various commercial samples from the official sampling of the CVUA Karlsruhe in Baden-Württemberg, Germany, initially obtained for microbiological analysis (see Section 2.5), were included. Besides PCA, visualization of spectra was conducted using quantile plots of both data sets (i.e., cold and hot brewed coffees), which are a useful tool for identifying changes within 1D ^1^H NMR datasets as color gradients.

### 2.4. Sensory Tests

Three ranking tests and two triangle tests were conducted at the Intergastra trade fair in Stuttgart, Germany, 15–19 February 2020. Rank order testing was conducted according to the specifications of ISO 8587:2010 [12]. Products were evaluated by determining the order of preference in a general hedonic consumer test. In each ranking test, four samples were assigned a rank by each tester. The testers were untrained participants. They were informed at the beginning about how the test would be conducted. Subsequently, the samples were tasted by the testers and the ranks assigned to the individual samples were entered into the answer form.

The samples for the first ranking test were four nitro cold brews, each prepared with a different type of coffee selected as the most typical, commercially available types:Arabica pulped natural Catuaí (Fazendas Dutra, Brazil);Arabica fully washed Catuaí (Finca Hamburgo, Mexico);Arabica S795 (Palthope Estate, India);Canephora SLN274 (Badra Estates, India).

The samples of the second and third ranking test consisted of Arabica and Canephora beans, respectively, prepared as follows:Cold brew;Nitro cold brew;Flash cooled coffee (cooled with ice);Hot brewed coffee (cooled at room temperature).

For cold brew coffee, 2 kg of coffee beans were weighed and coarsely ground. The ground coffee was put into a filter, which was placed in a cold brew system (Toddy, LLC, Loveland, CO, USA). Subsequently, 11.2 L of water at approx. 23 °C room temperature were added to the container. After a 17 h extraction time, the cold brew extract was poured into glass carafes via the outlet tap. These were stored in a refrigerator until further use. Before conducting the experiment, the cold brew was diluted with water in a 1:1 ratio.

For the production of nitro cold brew, nitrogen was added to the cold brew by a DP-25 nitro dispenser (Nitro DP, Bessenbach, Germany). The cold brew, prepared as above, was put into a plastic container, which was connected to a hose leading to the device. The finished nitro cold brew was withdrawn through a tap.

The flash cooled coffee was prepared by weighing 30 g of beans and grinding them finely. The coffee powder was placed in a filter coffee maker, producing 500 mL of hot coffee. A 1 L measuring cup was placed under the coffee maker, containing 500 g of crushed ice. As the coffee ran out of the coffee maker, it flowed into the measuring cup with the crushed ice and was thus cooled immediately.

For hot brewed coffee, 120 g of beans were weighed and finely ground. The coffee powder was then placed in a filter coffee machine, which was used to produce 2 L of hot coffee. The coffee flowed from the coffee machine directly into a thermos flask. This was left to cool with the lid open until the coffee had reached room temperature.

To keep the size of the statistical risks as small as possible, a number of participants of 60 was aimed for. A significance level of α = 0.05 was chosen. For the evaluation, the Friedman test according to ISO 8587:2010 [12] was applied.

The triangle tests were carried out according to the specifications of ISO 4120:2007 [13] to investigate whether there is a perceptible sensory difference between the samples. The testers were each given three samples, two of which were the same and one of which was different. They were instructed to indicate which sample they thought was different from the others.

The number of test persons in the test necessary for determining a difference was usually between 24 and 30 persons [13]. Therefore, a number of 25 persons was specified for carrying out the triangle test, as well as a significance level of α = 0.05. The minimum number of correct answers for α = 0.05 and 25 test persons has to be 13 [13].

### 2.5. Microbiological Survey of Cold Brew Coffees from the Market

During summer 2020, a total of 23 different cold brew coffee samples were collected from coffee shops, including those with an affiliated roasting house, in southern Germany. For details of the survey, see [14]. Sterile containers were used and the samples were transported at 8 °C to the laboratory. The samples were examined using international and German reference methods (DIN 10109:206-05, DIN 10164-1:2019-06, DIN EN ISO 13720:2010-12, DIN EN ISO 6888-2:2003-12, DIN EN ISO 7932:2020-11, BVL L 01.00-37:1991-12) for a wide range of microorganisms that have been reported as causing spoilage or health risks, such as aerobic lactic acid bacteria, yeasts, hygiene indicators such as Enterobacteriaceae, *Pseudomonas* spp. and coagulase-positive Staphylococcus, potentially pathogenic germs such as presumptive *Bacillus cereus* and the pathogens *Listeria monocytogenes* and *Salmonella* spp., as well as for molds.

## 3. Results

### 3.1. Results of the Online Survey on the Cold Brew Production Process

Seventy-one percent and thus most participants of the survey prepare cold brew at home. The second most frequent participants, at 22%, were people who prepare cold brew commercially on a small-scale basis and, in 7% of the cases, cold brew was produced industrially.

Table 1 shows that most of the participants have advanced knowledge, with 1 to 5 years of experience. Fifty-six percent of the respondents belong to this group. Among the commercial producers are the most experienced people (22%). The least experienced are the people who make cold brew at home.

Figure 1 shows that in private households, glass containers are mainly used. In the commercial sector, commercial systems such as the Toddy system are also used. In industry, stainless steel containers are mainly preferred. Overall, commercial systems are most commonly used to make cold brew. It should be noted, even though it is only a small percentage of 7%, some participants use hot brewed coffee for making cold brew.

Regarding the coffee–water ratio, 44% of participants use between 50 g and 100 g of coffee per 1 L of water to extract the cold brew, with only a few (5%) using less. However, some use more than 100 g per liter (36% in the range 100–150 g/L, 6% in the range 150–200 g/L and 6% use more than 200 g/L).

Most of the respondents were unable to specify their water composition. If this percentage is not taken into account, most of the respondents (18%) use soft water for cold brew extraction. For home brewing, most use untreated tap water (22%), commercial makers prefer soft water (35%) and industrial makers use medium–hard water (33%).

A temperature of 0 °C typically represents the cold drip method, 8 °C represents extraction temperature in the refrigerator, 20 °C and 30–40 °C are both used for room temperature extraction depending on the climate zone. According to Figure 2, most of the respondents extract cold brew at refrigerator temperatures of about 8 °C. This temperature is also mostly used for extractions at home, as well as in commercial operations. In industry, extraction is predominantly carried out at a room temperature of 20 °C.

From Figure 3, it can be seen that the majority of the respondents extract the cold brew for 14 to 26 h. The median extraction time is 16 h. Some also extract between 8 and 14 h, but few of the respondents extract for less than 8 h and even fewer extract for more than 26 h. In addition, it is notable that there is a small proportion who extract for more than 44 h.

As can be seen in Figure 4, coarsely ground coffee is mainly used in each group (home, commercial and industrial). Fine ground coffee is used the least.

Most of the respondents (53%) use Arabica beans for making cold brew coffee. Only a few (1.7%) use Canephora, Liberica or a blend of several species (3%). However, there are also some participants (41%) who do not know what kind of variety they use.

From Figure 5, it can be seen that medium-roasted beans are preferred by 54% of all respondents. Nineteen percent use a dark and 15% a light roast. An espresso roast is used by only four out of 123 people, and no one uses a very dark espresso roast.

Next, it was investigated how long the respondents keep the cold brew after extraction. On average, cold brew is kept for 1.5 days, and it can be seen from Table 2 that the maximum time that cold brew is kept is 7 days and the minimum is less than one day. The median for private and commercial groups is about half a day. In industry, the cold brew is kept for about a day (probably, in this case, this is the time before pouring into cans/sterilization).

Table 3 shows a summary of how the cold brew is served. By most participants, the cold brew is served with ice cubes, in a large glass, without milk, without sugar and not as nitro.

It was investigated at what temperature the beverage is at when it is served. It can be seen from Figure 6 that most serve their cold brew cold, mostly iced. Very few serve their cold brew warm, but some at room temperature or hot.

A final question allowed participants to describe what they particularly like about the cold brew extraction method compared to others. The respondents see the taste as the main advantage of cold brew, especially the smooth, less sour and less bitter taste. In addition, it was mentioned that the cold brew is refreshing and something different.

### 3.2. Results of Extraction Experiments

Figure 7 shows the results of the first cold brew extraction experiment. The figure shows that for formic acid, after 40 min, there are no longer any major changes in the content. When looking at chlorogenic acid, an increase can be seen up to 120 min. The caffeine content stagnates after about 140 min. The acetic acid content no longer shows any major changes after 40 min. Overall, no major changes can be seen in the HMF content. The lactic acid content stagnates after just 20 min. Trigonelline no longer increases significantly after 40 min.

Figure 8 and Table 4 compare the concentrations of the investigated ingredients in cold brew after extraction using different agitation methods. The extraction in the ultrasonic bath resulted in an average increase of 39%. The shaking table resulted in an average increase of 11%.

### 3.3. Discrimination between Cold and Hot Brews

Exploratory data analysis using PCA of full NMR spectra shows a separation of hot and cold brews (Figure 9). The quantile plots (Figure 10) as well as the loadings plots (data not shown) provide evidence that there are no single marker compounds, which would allow the differentiation between hot and cold brews, but that the differences are rather caused by minor intensity differences of the same resonances.

The classification with a partial least squares regression-discriminant analysis (PLS-DA) was also successful (data not shown). The resulting model showed acceptable predictive power in the preliminary validation performed. Nineteen out of 25 commercial samples were correctly classified.

### 3.4. Sensory Test Methods

#### 3.4.1. Ranking Test

To gain insights into the preference for differently prepared coffees, three different ranking tests were conducted, one with four different coffee varieties prepared as nitro cold brews and two with four different preparation/extraction types for the same Arabica or Canephora beans.

In the first trial, the ranking expressed as average ranking (*n* = 61) was as follows: Arabica pulped natural Catuaí (Fazendas Dutra) 2.7; Arabica fully washed Catuaí (Finca Hamburgo) 2.6; Arabica S795 (Palthope Estate) 2.5; and Canephora SLN274 (Badra Estates) 2.2. The null hypothesis that there are no differences between the samples cannot be rejected (F_test_ (4.44) < F (7.81)).

In the ranking trial with differently prepared Arabica coffees, the ranking expressed as average raking (*n* = 60) was as follows: cold brew 2.7, nitro cold brew 2.7, shock chilled coffee 2.4, hot brewed coffee 2.2. The null hypothesis cannot be rejected (F_test_ (6.03) < F (7.81)). That is, based on this result, there are no differences between the samples.

Finally, in the ranking trial with differently prepared Canephora coffees, the ranking expressed as average raking (*n* = 59) was as follows: nitro cold brew 2.7, cold brew 2.6, shock chilled coffee 2.4, hot brewed coffee 2.3. The null hypothesis cannot be rejected (F_test_ (3.57) < F (7.81)). That is, based on this result, there are no differences between the samples.

#### 3.4.2. Triangle Test

One test investigates whether there is a significant difference between hot brewed and cold brew coffee and the other whether there is a difference between two cold brews with Arabica fully washed Catuaí (Finca Hamburgo) and Arabica S795 (Palthope Estate).

The results of the triangle tests are shown in Table 5. In the cold brew vs. hot brew test, a large proportion of participants correctly identified the deviant sample. At a significance level of α = 0.001, the samples differ significantly from each other. Here, the cold brew is preferred by 17 participants, 6 prefer the hot brewed coffee and 2 of the participants had no preference. On the other hand, in the trial with cold brews from different coffee varieties, only 12 of the 25 participants correctly identified the deviant sample. This means that the samples are not significantly different at a significance level of α = 0.05. Fourteen testers prefer the Arabica fully washed Catuaí (Finca Hamburgo) and 10 prefer the Arabica S795 (Palthope Estate), one tester has no preference.

### 3.5. Microbiology Results

An increased microbial load was found in two samples (9%). One of the samples showed a clearly increased bacterial count of potential spoilage organisms, namely lactic acid bacteria and yeasts. In another sample, contamination with presumptive *Bacillus cereus* was detected. This was a cold brew coffee with a storage time of five days.

Two forms of gastrointestinal diseases (emetic and diarrheal syndrome) caused by *Bacillus cereus* are known. These microorganisms are opportunistic food-borne pathogens, producing several toxins that have been associated with food poisoning, though mostly corresponding with a considerably higher microbial load than what was found in the samples.

## 4. Discussion

### 4.1. What Is a Cold Brew Coffee?

Table 6 summarizes the most frequently chosen survey responses.

The results of the survey show that cold brew is still rather new on the market and there are much fewer companies in the industry that produce and sell cold brew compared to hot brewed coffee. This is also confirmed by the question about experience, as this is between one and five years for most respondents. The responses about the extraction methods show that it is necessary to define what a cold brew is. Some still refer to hot extracted coffee, which has subsequently been cooled, as cold brew, which is also evident from the question about extraction temperature. The question on dosage shows that there is no agreement, as the values scatter over a wide range. The same applies to water quality, although soft water is preferred. When it comes to the grinding degree, however, most agree that the beans should not be ground finely. Too fine a grind can cause the coffee to be too intense [16]. About 6% of the respondents extract the coffee for more than 44 h. This cold brew may be kept for up to 7 days. In addition, 41% extract the cold brew at 20 °C and above. Especially from a microbiological point of view, it is important to determine a maximum time and temperature for both extraction and storage conditions. If coffee is not brewed hot and then stored at room temperature for hours or even days, this favors the growth of microorganisms including some pathogens [17]. The microbiological testing results (see Section 4.4) corroborate that there is more than a hypothetical hazard but rather a clear risk during cold brew coffee processing. As long as it has not been empirically determined up to which time and under which conditions the cold brew is considered safe, however, it is recommended not to keep it longer than one day and to discard it immediately if changes in the smell or taste become noticeable [8].

There appears to be a consensus, however, on the choice of bean type. Most prefer Arabica over Canephora for making cold brew coffee. This might be caused by the typical marketing information about a putative higher quality of “100% Arabica” and the actually inferior quality of most commercially available Canephora.

A medium roast degree is often used for roasting. However, some roast light or dark. As mentioned above, this has a significant impact on the flavor of the cold brew, especially depending on the degree of defective beans.

Cold brew coffee can also be varied in terms of serving temperature. Most people are not yet aware that it can also be consumed hot. This offer is not yet very well represented on the market.

### 4.2. Insights on Cold Brew Coffee Extraction

The actual amount of components in coffee depends on factors such as the bean, processing, chemical composition, roasting process and extraction method. For example, in cold brew, low extraction temperatures and longer extraction times provide different physicochemical and sensory properties. Thus, cold brew coffee has a different aroma profile and flavor profile compared to hot brewed coffee [8]. Cold brew coffee may be served by adding nitrogen, so that the resulting froth not only changes the feel in the mouth, but also provides a more intense flavor by increasing the surface area [8]. The enrichment with nitrogen is usually done with a nitro dispenser with a built-in compressor. The nitrogen required may be extracted from the air (e.g., using a specific dispensing device as used in the sensory experiments).

Turbulence and the temperature of the water have the most decisive influences on the extraction of the investigated substances. With increasing turbulence and temperature, the extraction rate increases.

The course of the extraction without agitation behaved according to the expectation that the extraction would increase slowly and steadily after an initial, rapid increase. These assumptions were based on the fact that after the initial strong shaking for homogenization, the diffusion of the ingredients of the coffee powder into the water would be strongly slowed down. This was due to the fact that the contact area between the water and the coffee powder was greatly reduced as the powder settled to the bottom of the bottle.

During extraction in the ultrasonic bath, the initial increase in concentration was much less pronounced compared to the rest of the extraction process. This was followed by a steady and much steeper increase in concentration compared to extraction without agitation.

Our results suggest that at low turbulence and an extraction temperature of about 23 °C, an extraction time of more than 7 h is unnecessary; at higher turbulence, an extraction time of 2 to 3 h is already sufficient. If extraction is carried out at colder temperatures in the refrigerator, not all substances will be completely dissolved after 6 h at low turbulence.

In this work, no changes are apparent after more than 7 h, but several studies report changes at longer extraction times [18,19]. On the one hand, this could be due to the fact that additional substances are extracted, which were not measured in this study, or due to the different parameters of the extraction, such as temperature, dosage, turbulence, etc.

Furthermore, the question must be clarified as to what percentage should actually be extracted in order to produce a high-quality cold brew. A very long extraction time for cold brew has always been considered positive [20], however, the study by Han et al. [19] showed the opposite in a sensory test. As with hot extraction methods, e.g., espresso, not all soluble components should be extracted, but only enough to extract all desired flavor compounds and not the undesired ones. In this process, the risk of contamination should also be kept as low as possible [21]. Nevertheless, the typically long extraction time of about one day could have to do with the fact that in the daily routine it is simply practical to schedule the preparation of cold brew the day before for the following day.

A final comparison of the extraction methods investigated here shows that a cold brew extraction is accelerated the most by ultrasonication. Other authors suggested extraction under reduced pressure as the best treatment for the acceleration of cold brew coffee extraction [22]. However, this technique might be technologically more demanding than ultrasonication. Another parameter that influences or increases the extraction is the particle size, with medium sizes (0.7–1.7 mm) suggested as accelerating the extraction rates [23]. Kwok and colleagues suggest an extraction of 70% of the extractable fractions as optimal [8]. This assumption still needs to be verified using further sensory testing. A recent study also hypothesized a non-linear extraction behavior [24], which was not observable in our investigations, but clearly needs further testing.

### 4.3. Analytical Discrimination of Cold from Hot Brewed Coffee

During NMR analysis, there was not any distinct signal that would be observable only in cold or hot brewed coffee, but the spectral differences are nuanced, so a multivariate method is necessary for data analysis. Initial PCA of the hot and cold brews from the same beans was successful, as there was no overlap of the clusters. This successful separation gives a first indication that establishing a model for checking and validating labeling claims of cold brew coffee could be possible.

Thus, of the 25 commercial samples whose production method was largely unknown (apart from the manufacturer’s labeling), 19 were correctly classified. The remaining six samples can be explained by either mislabeling or being outside of both model spaces (e.g., by using other ingredients such as milk and/or sugar). Some of these mixed drink samples also had some spectral problems such as comparably larger half-widths of NMR signals, so that a methodological improvement needs to be done for samples containing more ingredients than a coffee/water mixture.

Nevertheless, these results suggest that validation of the authenticity of cold brews using a multivariate model may be possible in the future. However, for this to happen, the model needs to be strengthened by a large number of additional samples including diverse factors such as degree of roasting, type of coffee, degree of extraction, etc. This should improve the predictive power and confidence of the classification.

### 4.4. Sensory Properties of Cold Brew Coffee

In the first ranking test, in which nitro cold brews prepared with different coffee beans were compared, there is no significant difference between the samples. This could be due to the fact that the participants are untrained testers, for whom the difference between the varieties is too small and they therefore do not perceive it or hardly perceive it. Nevertheless, there is an indication that the Arabica pulped natural coffee is the most popular. This is the sweetest coffee, while the second in the order, Arabica fully washed, had the most citrus character. This finding is also in line with the research of Seninde et al. [25] based on a highly trained sensory panel, which suggested Canephora cold brews as being more bitter, while wet-processed Arabica was most associated with being balanced, and natural processed Arabica was most differentiated by fruity, floral, fermented and sweet aromatics.

The rank order test with the different preparation types also shows no significant differences, neither for the Arabicas nor for the Canephoras. This is probably due to the fact that high-quality specialty coffee was applied, making even the cooled-down hot brew coffee drinkable.

Since taster expectation has a role in assigning rank numbers, nitro cold brew, visually clearly distinguishable, therefore probably scores higher [26]. This is in contrast to the consumer acceptance testing results of McCain-Keefer et al. [27], where nitro cold brew coffees scored significantly lower than other cold brews in overall liking. However, the samples were commercial products and not necessarily based on the same coffee and extraction parameters, so these results are not directly comparable.

The triangle test of hot brewed and cold brew coffee shows that the two preparation types are distinguishable from each other with a high probability. Here, the cold brew coffee was preferred. This supports the results of the ranking test, in which cold brew was preferred over hot brewed coffee. Our results also corroborate previous literature about considerable taste differences between hot and cold brews [28], which may be caused by differential extraction of key characteristic compounds such as acids [29] and several volatile compounds [30].

However, the two Arabica varieties of Arabica fully washed Catuaí (Finca Hamburgo) and Arabica S795 (Palthope Estate) could not be distinguished from each other in the triangle test. The results indicate that the testers may lack experience and knowledge. This suggests that the difference between the tested Arabica varieties is too small to be detectable by a consumer.

### 4.5. Hygiene Requirements for Cold Brew Manufacture

Coffee is exposed to many microbial hazards during processing and production. The warm, humid and tropical climate of the coffee cultivating areas provides an ideal environment for the growth of various species of fungi. Processing and storage may lead to further microbial contamination, especially with *Enterobacteriaceae* [14].

While the roasting of coffee beans, the use of very hot water and the immediate consumption at hot temperatures typically lead to a negligible microbial contamination of hot brewed coffee, cold brew coffee can be classified as a beverage that requires special hygiene requirements in terms of food contamination, i.e., food safety. This is due to the fact that no heating process takes place during the production of cold brew coffee and thus there are no microbicidal effects [8,14].

Cold brew coffee is a slightly acidic environment (pH 4.9–6.0) which does not inhibit microbial growth [14]. Consequently, yeasts, molds and lactic and acetic acid-producing bacteria can multiply during the long extraction process and the subsequent storage [8]. Besides the spoilage agents mentioned, pathogenic germs such as *Salmonella* or *Listeria* must be taken into account. Possible sources of contamination should be identified to ensure the microbial stability of cold brew coffee. For example, the equipment, containers and ingredients used as well as the personnel may contribute to microbial contamination and thus compromise food safety. Therefore, cold brew coffee must receive special consideration in the hazard analysis critical control point (HACCP) concept [14].

## 5. Conclusions

Based on the laboratory evaluations, it can be concluded that an extraction time of more than 7 h is typically not necessary to prepare a cold brew coffee. At high turbulence, extraction maxima can be as low as 2 to 3 h. The results of the tests also show that as turbulence and temperature increase, the extraction rate increases. In addition, the sensory tests indicate that the type of coffee has an influence on the sensory properties of the cold brews. They also indicate that cold brew is preferred to a hot extracted coffee, which has been cooled, so a hot extracted coffee should never be called a cold brew.

However, there is no such thing as the optimal method of production, as the optimal cold brew varies according to personal preference. What should be established is that cold brew coffee is extracted with cold water and what is meant by cold. In addition, it should be known which parameters have to be changed and how, so that the desired sensory profile becomes noticeable. Therefore, further experiments should be carried out which investigate the influence of the bean, the grind, the water composition, the extraction vessel, the storage time of the beans and the roasting. It would also be interesting to investigate other substances in the coffee when carrying out the experiments. For the extraction, it is also important to determine what proportion of extracted substances is optimal for a cold brew. It is also important to determine what microbial risks cold brew presents and what extraction and storage times can be classified as safe. In addition, further sensory tests should be carried out, which determine which extraction time, roasting, etc. are preferred.

Fortunately, only a small proportion of the samples examined showed microbial contamination. However, the survey has shown that the compliance with the hygiene requirements for the production of cold brew coffee should be regularly monitored. The risk of microbial contamination of cold brew coffee may be compared to that of non-alcoholic beverages on draft from beverage dispensing systems. To identify a possible source of contamination, a step-by-step approach is necessary with particular attention to the ingredients and the extraction process, as well as to the storage conditions of the final product.

In general, it is recommended that cold brew coffee is freshly made and consumed the same day. A long storage period (several days to weeks) of cold brew coffee leads to an increased risk of microbial contamination and this affects the taste. In this context, cold brew coffee should be compared with filter coffee: Filtered coffee would never be stored for such a long time but rather be discarded after a few hours due to its stale taste. The same should be done with cold brew coffee at the end of the working day unless microbial safety and product quality are otherwise ensured.

## Figures and Tables

**Figure 1 foods-10-00865-f001:**
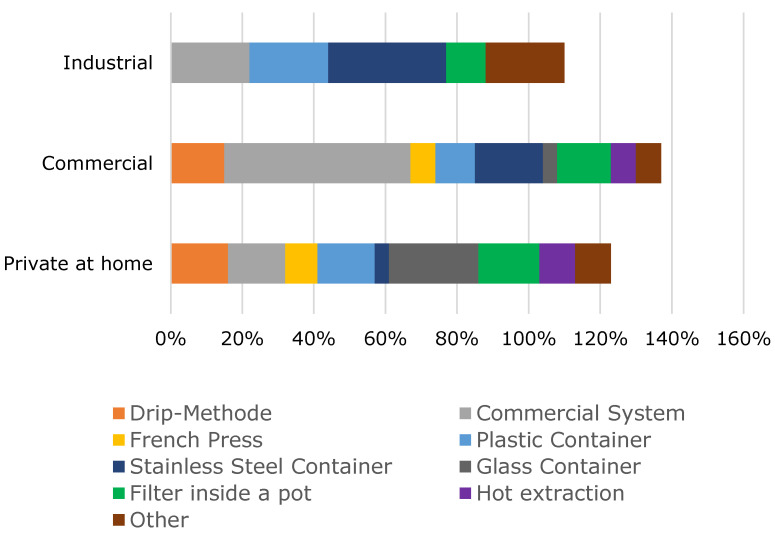
Extraction method of cold brew coffee (note: percentage values above 100 to be explained by multiple answers).

**Figure 2 foods-10-00865-f002:**
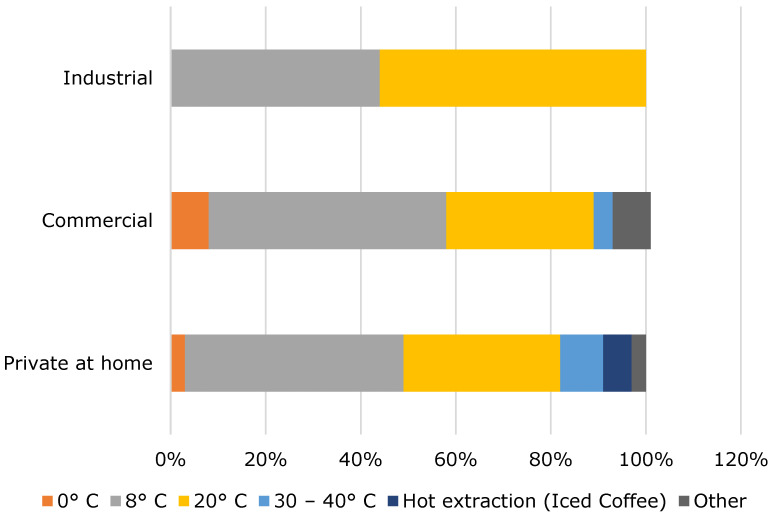
Extraction temperature of cold brew coffee.

**Figure 3 foods-10-00865-f003:**
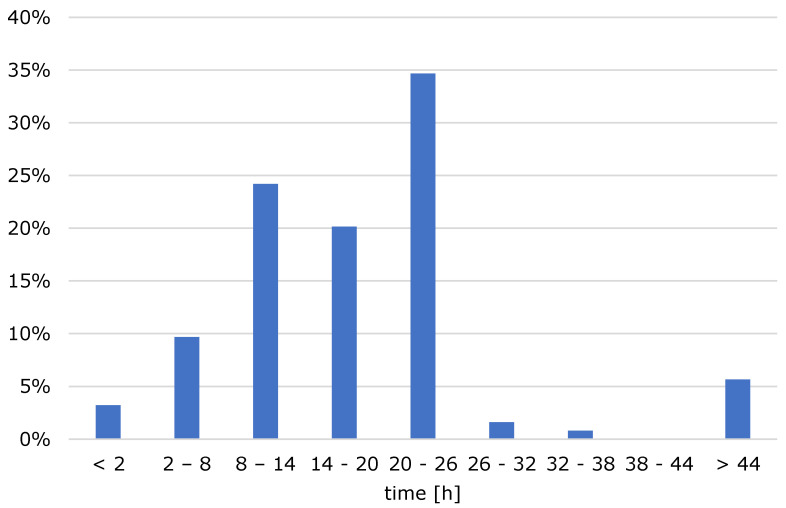
Extraction time of cold brew coffee.

**Figure 4 foods-10-00865-f004:**
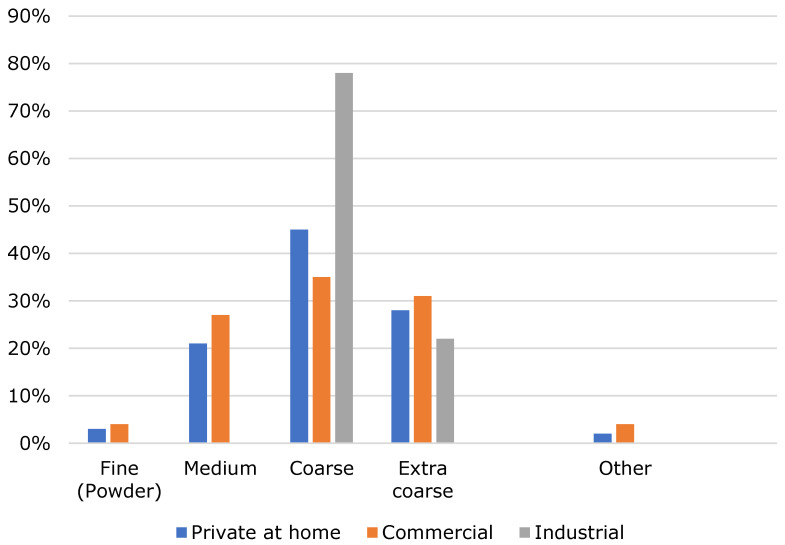
Grinding degree of cold brew coffee.

**Figure 5 foods-10-00865-f005:**
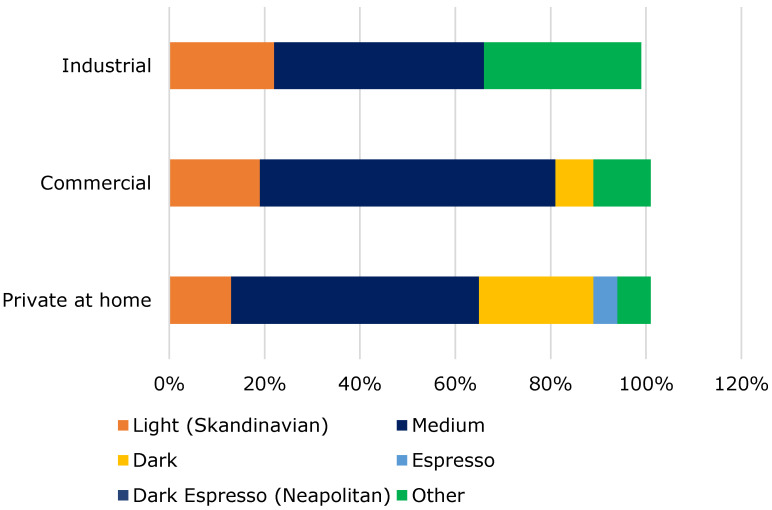
Roasting for preparing cold brew coffee.

**Figure 6 foods-10-00865-f006:**
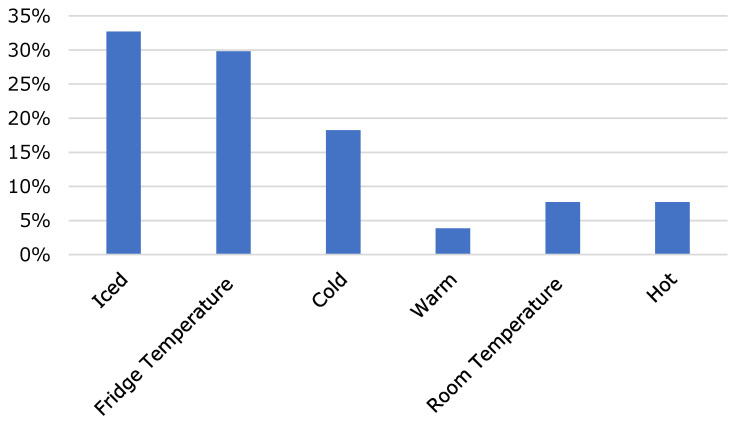
Consumption temperature of cold brew coffee (temperatures up to 4 °C: “Iced”; 5 to 8 °C: “Fridge Temperature”; 9 to 12 °C: “Cold”; 13 to 15 °C: “Warm”; 16 to 33 °C: “Room Temperature”; temperatures above this: “Hot”).

**Figure 7 foods-10-00865-f007:**
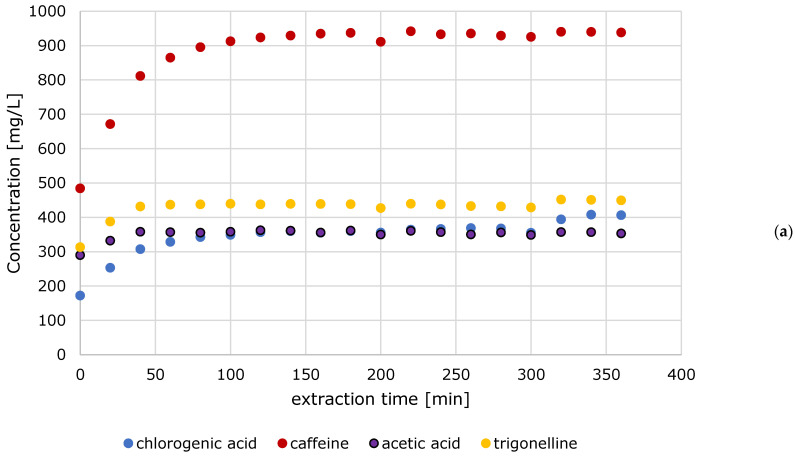
Evolution of chlorogenic acid, caffeine, acetic acid, trigonelline (**a**) and formic acid, 5-hydroxymethylfurfural (HMF), lactic acid (**b**) (*n* = 1).

**Figure 8 foods-10-00865-f008:**
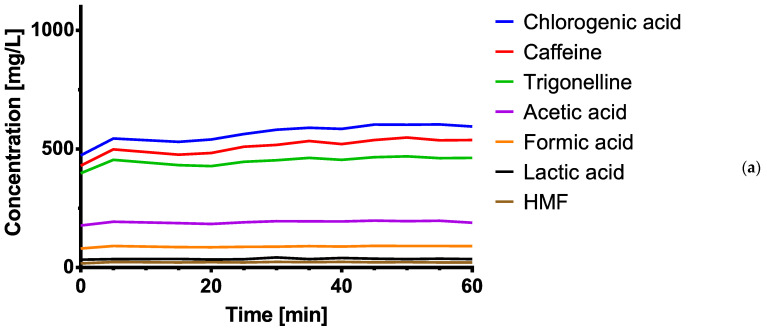
Graphical representation of the extraction process measuring points every 5 min: (**a**) without agitation; (**b**) with ultrasonication; (**c**) with constant agitation (*n* = 1).

**Figure 9 foods-10-00865-f009:**
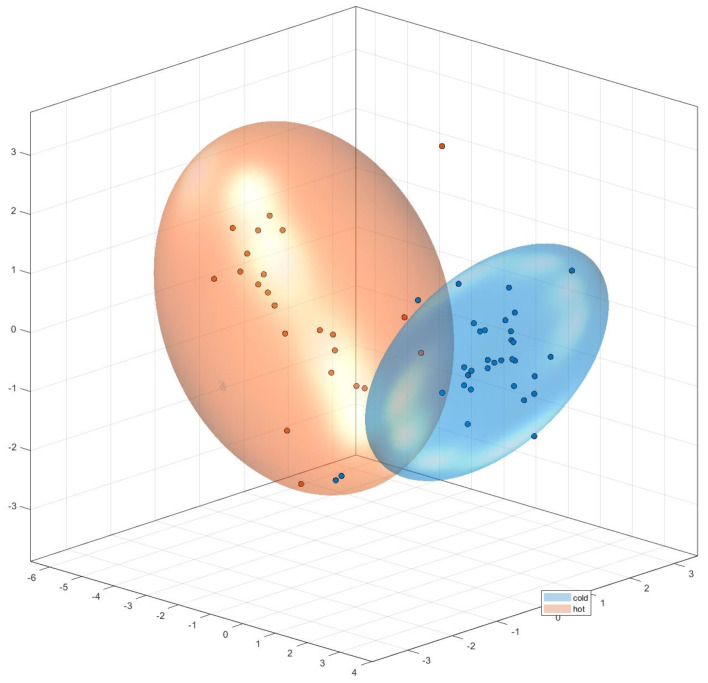
Principal component analysis (PCA) of hot (red) and cold (blue) brew coffees.

**Figure 10 foods-10-00865-f010:**
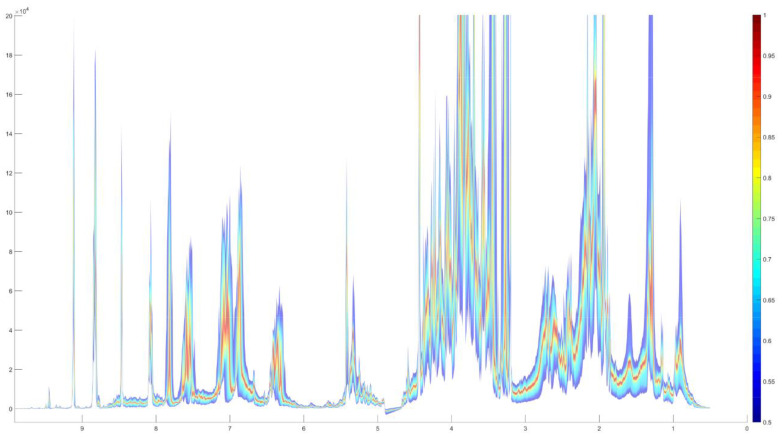
Quantile plots of NMR spectra of hot (upper panel) and cold brew coffees (lower panel).

**Table 1 foods-10-00865-t001:** Experience with the production of cold brew coffee.

	Rather New (<1 Year)		Intermediate (1–5 Years)		Experienced (>5 Years)	
Private at home	35%	31	53%	47	12%	11
Commercial	15%	4	63%	17	22%	6
Industrial	22%	2	67%	6	11%	1
Total Responses	29%	37	56%	71	14%	18

**Table 2 foods-10-00865-t002:** Storage time of prepared cold brew coffee (in days).

	Private at Home	Commercial	Industrial
Minimum	0	0	0
Maximum	7	6.3	7
Median	0.4	0.5	1.3
Mean	0.8	1.0	2.6
Standard Deviation	1.4	1.4	2.6

**Table 3 foods-10-00865-t003:** Survey result of the different methods to serve cold brew coffee.

Temperature		Vessel Size		Milk		Sugar		Nitro	
Iced	33%	Standard glass/mug	36%	No	53%	No	67%	No	74%
Fridge Temperature	30%	Large glass	49%	Yes	18%	Yes	12%	Yes	14%
Cold	18%	Small shot glass	15%	Sometimes	20%	Sometimes	14%	Sometimes	13%
Warm	4%			Non-dairy	9%	Syrup or honey	6%		
Room Temperature	8%								
Hot	8%								

**Table 4 foods-10-00865-t004:** Percentage increase in the amount of substance after a one-hour cold brew extraction with the specified method compared to an extraction without agitation.

	Formic Acid	Chlorogenic Acid	Caffeine	Acetic Acid	HMF	Lactic Acid	Trigonelline
Ultrasonication	+16%	+71%	+26%	+21%	+16%	+81%	+19%
Constant agitation	+3%	+26%	+10%	+5%	+13%	+21%	+3%

**Table 5 foods-10-00865-t005:** Results of the ISO 4120:2004 sensory analysis using triangle testing for differentiation of cold brew coffee.

Test Material	No. of Assessors	No. of Correct Responses	Significance ^1^	LCI/UCI ^2^
Hot brew vs. cold brew	25	20	yes (α = 0.001)	0.50/0.90
Nitro cold brew: Arabica Catuaí (Brazil) vs. Arabica S795 (India)	25	12	no	- ^3^

^1^ According to ISO 4120:2004 [13]. For the non-significant trial, the minimum number of correct answers to conclude that a perceptible difference exists (α = 0.05) would have been 13/25. ^2^ Lower and upper 95% confidence intervals (LCI/UCI) for the triangle tests calculated according to ISO 4120:2004 [13]. The limits can be interpreted as percentage of population that can perceive a difference between the samples [15]. ^3^ Not calculated for non-significant trial.

**Table 6 foods-10-00865-t006:** Summary of most typical conditions for cold brew extraction.

Parameter	Most Frequent Response	Percentage of Response
Preparation site	at home	71%
Experience	1–5 years	56%
Extraction method	commercial system	19%
Dosage	50–100 g/L	44%
Water hardness	soft	44%
Extraction temperature	8 °C	47%
Extraction time	20–26 h	35%
Grinding degree	coarse	45%
Coffee type	Arabica	53%
Degree of roasting	medium	54%
Storage time	1.5 days	Average
Preparation	- with ice cubes - in a large glass - without milk - without sugar - without nitrogen	
Advantages	less acidity and bitterness	

## Data Availability

The questionnaire is available online at ResearchGate, doi:10.13140/RG.2.2.19212.67206/1. The survey data (anonymized results) presented in this study are openly available at ResearchGate, doi:10.13140/RG.2.2.23342.95044.

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
