# Peer review of "Cold Brew Coffee—Pilot Studies on Definition, Extraction, Consumer Preference, Chemical Characterization and Microbiological Hazards"

_foods, 2021, doi:10.3390/foods10040865_

Round 1
Reviewer 1 Report
It is a very comprehensive article with a very broad objective.
Some comments or suggestions:
Justification of the selected varieties may be required.
Coffee grinding is well represented, but characterized by the mean particle diameter would improve it.
Figures 7 and 8 show that, in general, the extraction is very fast, which could suggest that the procedure could be modified taking this result into account.
Author Response
It is a very comprehensive article with a very broad objective.
Thank you!
Some comments or suggestions:
Justification of the selected varieties may be required.
The selection was based on the most typical varieties that make up the market. The major aim was to investigate a product that is also relevant for the large coffee market. Hence, we chose common varieties of Arabica and Canephora. The selected Indian Canephora and the red Catuai are far apart both from origin and genetics, but also preference. On the other hand, we did not want to select extremes, but medium qualities that really exist in large volumes, but of course, still in the form of a top high-end selection with absence of defects. Therefore, we did not include inexpensive Vietnamese mass coffee where you can already taste out the defects. The selection then included aspects of fruitiness and acidity. Therefore, we selected one Arabica where you have a high acidity. There may certainly be a preference for the gently sweet Arabica, and also over the fully-washed one which has less sweetness.
Coffee grinding is well represented, but characterized by the mean particle diameter would improve it.
The manual of the used mill Mahlkönig EK43 does not provide a specification. However, a detailed particle size and grind quality assessment was published on the internet: Thomas Manion. Quantitative Assessment Of Mahlkonig EK-43 Burr Alignment. https://flightcoffeeco.com/blogs/news/our-mahlkonig-ek-43s-grind-quality
According to that, the grind size 8, which we have applied, has an average diameter of 0.94 mm.
Figures 7 and 8 show that, in general, the extraction is very fast, which could suggest that the procedure could be modified taking this result into account.
Thank you, we fully agree and this is actually one of our main conclusions that the procedure that is current practice may be much too long.
Reviewer 2 Report
Similar extraction experiments have been reported previously. However, it is unclear that the authors are aware of the previously published research. The scope of the manuscript is too broad. Each section is disjointed from others. I would recommend that the authors resubmit the manuscript with a narrowed focus.
Author Response
Similar extraction experiments have been reported previously. However, it is unclear that the authors are aware of the previously published research.
We are aware of previous experiments and have cited them. However, we actually do not believe that previous experiments really provided insights into the kinetics of cold brew extractions starting from minute 0. They typically compared very long times such as 18 to 24 hours.
The scope of the manuscript is too broad. Each section is disjointed from others. I would recommend that the authors resubmit the manuscript with a narrowed focus.
The authors disagree that research needs to presented in the “smallest publishable unit” as it is sometimes typical practice, often exacerbated by low word-limits of traditional journals. As Open Access Publishers such as MDPI do not enforce word-limits and allow fully-described research papers, we took the chance to study all aspects of cold brew in this single paper. We actually do not see any merit to split this up into several papers.
Reviewer 3 Report
The authors should describe participant age, sex and other profiles.
Figure 6. does not require decimal point.
Figure 7, require Sd (standard deviation) from replications and statistical analysis.
For NMR data, it is not clear how many components were identified by NMR and from the PCA analysis, loading plots and significantly different metabolites should be mentioned and discuseed more.
Table 4 need statistical analysis.
Figure 9 is good information but biomarker that related with discriminant feature can be displayed in manuscipt.
There is not description how the author detect molds and bacteria.
2.5. Microbiological Survey of Cold Brew Coffees from the Market should be carefully rewritten.
Author Response
The authors should describe participant age, sex and other profiles.
We tried to collect demographic data with the questionnaires. However, many participants decided not to fill out this non-mandatory data. E.g., in the first triangle test there were 13 males, 7 females, 5 without data (age range for the ones with data 21-63). In the second triangle test there were 11 male, 8 female and 6 without data (age range 21-63). As an analysis of sub-groups is not possible, we believe this data would not be meaningful and decided not to include it in the paper.
Figure 6. does not require decimal point.
Thank you. The decimals were removed from the y-axis.
Figure 7, require Sd (standard deviation) from replications and statistical analysis.
As each NMR analysis is rather expensive, we decided that it would be more worthwhile to have more different time points than replicates of the same time point (which due to NMR’s exceptional precision would more or less lead to the same point where the SD is so small that it would not be visible in the diagram). We added information to the legend for clarification.
For NMR data, it is not clear how many components were identified by NMR and from the PCA analysis, loading plots and significantly different metabolites should be mentioned and discussed more.
We were unable to identify single components that would lead to differentiation. The differences are rather caused by different intensities of the same signals.
Table 4 need statistical analysis.
The table shows a percentage difference between treatments. We did not conduct statistical analysis and do not make claims on probabilities.
Figure 9 is good information but biomarker that related with discriminant feature can be displayed in manuscript.
We have added another figure showing the quantile plots of cold brews and hot brews for comparison. As stated above, it was impossible to find a single metabolite responsible for a differentiation. The discrimination is caused by changes in intensities.
There is not description how the author detects molds and bacteria.
The methods were standard norm method using colony enumeration. We have added the references to the norms in the text. As the matrix “cold brew coffee” is not included in the norms, we have plated the matrix similar to other foods and beverages.
2.5. Microbiological Survey of Cold Brew Coffees from the Market should be carefully rewritten.
See above, we have added references but do not see much merit in describing standard methods again. The novelty in this paper is that microbiological testing was done with the matrix of cold brew coffee for the first time in the literature but not the methodology itself.
Reviewer 4 Report
Despite of thefact thet is is rather essay than typical paper it needs detailed presentation of results in supplementary part..
- Half of the paper is devoted to the reults of a survey demonstrating how cold brew coffe is prepared. In this respects questionarre should be presented. There is a reference to the previous paper, however I did not find such questionnare there.
- Results of PCA analysis is not clear - which variables are considered in order to perform it?
- The analysis of content of chosen compounds in brewed coffe was done on the basis of NMR. Representative spectra should be shown to visualise the biggest differences observed. Did Authors use internal reference compounds there
All of these data could be included as Supplementary section without harming the major body of the manuscript.
Author Response
Despite of the fact that it is rather essay than typical paper it needs detailed presentation of results in supplementary part.
A supplement was already submitted with the initial submission of the paper (Excel table with full results). The file is publicly available at ResearchGate, doi:10.13140/RG.2.2.23342.95044 (direct link: https://www.researchgate.net/publication/348811901_Survey_results_cold_brew_coffee_-_raw_data_anonymizedxlsx).
Half of the paper is devoted to the results of a survey demonstrating how cold brew coffee is prepared. In this respects questionnaire should be presented. There is a reference to the previous paper, however I did not find such questionnaire there.
The questionnaire is available online at ResearchGate, doi:10.13140/RG.2.2.19212.67206/1. This is already stated at the bottom of the paper under the heading “Supplementary materials”. The DOI is active and working (direct link: https://www.researchgate.net/publication/340739045_Cold-Brew_Coffee_Preparation_Survey).
Results of PCA analysis is not clear - which variables are considered in order to perform it?
We did not conduct variable selection but used the full spectral information (see new figures with NMR spectra). We added “full NMR spectra” to the text to clarify this aspect.
The analysis of content of chosen compounds in brewed coffee was done on the basis of NMR. Representative spectra should be shown to visualise the biggest differences observed. Did Authors use internal reference compounds there
Representative spectra as quantile plots are now presented in a new figure of cold brew and hot brew coffees. Thank you for the advice. We did not apply internal reference compounds. This is unnecessary for NMR. For quantification we used the PULCON-ERETIC method, see Reference 9 (Okaru et al.).
All of these data could be included as Supplementary section without harming the major body of the manuscript.
We decided to include the spectra into the main body as this was also demanded by reviewer #3.
Round 2
Reviewer 3 Report
In Figure 1 2 and others, decimal points of Mean and SD are different. Please make them consistent. Use the same # of the decimal point.
Author Response
In Figure 1 2 and others, decimal points of Mean and SD are different. Please make them consistent. Use the same # of the decimal point.
ANSWER: We carefully re-checked all number presentations, especially in Figure 1 and 2, but did not identify any issue. All decimals we show are backed up by our method validation data. We do not present any non-significant decimals. It can well be that different methods or analytes have different significant decimals. Hence, it can be inappropriate to use the same number of decimals for all data points (e.g., suggest a higher level of accuracy by rounding replicates to more decimals than in the original measurement uncertainty).
Reviewer 4 Report
Now , after explanaitions and changes paper well revised.
In fact it is really interesting paper, however, system which authors used to present Supplementary data (reference to Research Gate to two intedepndent, not published papers in two after the manuscript) caused that I have overlooked them (possibly other readers will do the same). It would be more convenient to cite them just in text but I do not insist on that.
Author Response
Now, after explanations and changes paper well revised.
In fact it is really interesting paper, however, system which authors used to present Supplementary data (reference to Research Gate to two interdependent, not published papers in two after the manuscript) caused that I have overlooked them (possibly other readers will do the same). It would be more convenient to cite them just in text but I do not insist on that.
ANSWER: The supplementary data is cited as it is mandatory by the authors guideline and journal template. However, we are open to the suggestion to host the files at MDPI rather than at Research Gate. A short statement that the full raw data is available was added to the methods section.